# Cluster randomised controlled trial of screening for atrial fibrillation in people aged 70 years and over to reduce stroke: protocol for the pilot study for the SAFER trial

Kate Williams,[1] Rakesh Narendra Modi ![ORCID],[1] Andrew Dymond,[1] Sarah Hoare ![ORCID],[2] Alison Powell,[2] Jenni Burt,[2] Duncan Edwards,[1] Jenny Lund,[1] Rachel Johnson,[3] Trudie Lobban,[4] Mark Lown ![ORCID],[5] Michael J Sweeting,[6] H Thom,[3] Stephen Kaptoge,[7] Francesco Fusco,[1] Stephen Morris ![ORCID],[1] Gregory Lip ![ORCID],[8,9] Natalie Armstrong ![ORCID],[6] Martin R Cowie ![ORCID],[10,11,12] David A Fitzmaurice,[13] Ben Freedman ![ORCID],[14] Simon J Griffin,[1,15] Stephen Sutton,[1] FD Richard Hobbs,[16] Richard J McManus ![ORCID],[16] Jonathan Mant,[1] The SAFER Authorship Group[1]

For numbered affiliations see end of article.

**Correspondence to**
Dr Rakesh Narendra Modi;
rnm30@medschl.cam.ac.uk

## ABSTRACT

**Introduction** Atrial fibrillation (AF) is a common arrhythmia associated with 30% of strokes, as well as other cardiovascular disease, dementia and death. AF meets many criteria for screening, but there is limited evidence that AF screening reduces stroke. Consequently, no countries recommend national screening programmes for AF. The Screening for Atrial Fibrillation with ECG to Reduce stroke (SAFER) trial aims to determine whether screening for AF is effective at reducing risk of stroke. The aim of the pilot study is to assess feasibility of the main trial and inform implementation of screening and trial procedures.

**Methods and analysis** SAFER is planned to be a pragmatic randomised controlled trial (RCT) of over 100 000 participants aged 70 years and over, not on long-term anticoagulation therapy at baseline, with an average follow-up of 5 years. Participants are asked to record four traces every day for 3 weeks on a hand-held single-lead ECG device. Cardiologists remotely confirm episodes of AF identified by the device algorithm, and general practitioners follow-up with anticoagulation as appropriate. The pilot study is a cluster RCT in 36 UK general practices, randomised 2:1 control to intervention, recruiting approximately 12 600 participants. Pilot study outcomes include AF detection rate, anticoagulation uptake and other parameters to incorporate into sample size calculations for the main trial. Questionnaires sent to a sample of participants will assess impact of screening on psychological health. Process evaluation and qualitative studies will underpin implementation of screening during the main trial. An economic evaluation using the pilot data will confirm whether it is plausible that screening might be cost-effective.

**Ethics and dissemination** The London—Central Research Ethics Committee (19/LO/1597) and Confidentiality Advisory Group (19/CAG/0226) provided ethical approval. Dissemination will be via publications, patient-friendly summaries, reports and engagement with the UK National Screening Committee.

**Trial registration number** ISRCTN72104369.

## STRENGTHS AND LIMITATIONS OF THIS STUDY

⇒ Screening for Atrial Fibrillation with ECG to Reduce stroke (SAFER) is a large multicentre pragmatic randomised controlled trial planned to be the largest trial of atrial fibrillation (AF) screening that has been performed.
⇒ This internal pilot study will have good external validity, providing data on parameters for an AF screening programme in real-world conditions.
⇒ The process evaluation of the pilot study will inform the implementation of a large-scale AF screening programme.
⇒ Participant recruitment prior to cluster randomisation will ensure that intervention and control participants are similar, and are likely to take up screening if offered it.
⇒ Despite the fact that anticoagulation is indicated in some people under 70, the SAFER trial is not screening in this age group.

## INTRODUCTION

Atrial fibrillation (AF) is a cardiac arrhythmia present in approximately 10% of people aged over 65 years.[1] AF is increasing in prevalence,[2] and is associated with a fivefold increase in the risk of stroke,[3] as well as other negative health outcomes (such as heart failure, dementia and death).[4–8] While 30% of strokes are associated with AF, 10% of strokes occur

in people unaware that they have AF because it can be asymptomatic, intermittent ('paroxysmal AF') and/ or undiagnosed.[9–12] AF-related strokes tend to be more severe than strokes due to other causes, imposing burdens on patient, family, and health and social care systems.[10 13]

AF is diagnosed on an ECG.[14 15] This has traditionally been achieved by a health professional interpreting a 12-lead ECG. However, 30 s on a single-lead ECG is now regarded as sufficient to diagnose AF.[16–20] Furthermore, acceptable and accessible portable technologies such as wearable patches, smart watches and hand-held devices are available that can test for AF repeatedly over longer periods of time.[21 22] These technologies are sensitive to AF,[23] and can detect paroxysmal AF.[21 24]

Treatment with oral anticoagulation can effectively[25–27] and cost-effectively[28 29] reduce risk of stroke associated with AF, especially when part of an integrated care or holistic approach to AF management, as advocated in guidelines.[30 31] However, a sizeable minority of eligible patients are not taking anticoagulants.[21 32–36] With non-Vitamin-K antagonist oral anticoagulants (NOACs; also called direct oral anticoagulants) that require substantially less monitoring, and stronger recommendations for anticoagulation in clinical guidelines,[19 31] the rates of anticoagulation are increasing, but remain suboptimal.[37–39]

Undiagnosed AF is common and can be detected with simple and portable technology, and there are effective treatments available.[18 40–42] AF screening, therefore, fulfils many of the criteria for initiating a national systematic screening programme.[21 40 43 44]

However, no countries endorse national AF screening programmes.[14 31 33 45] Until recently, there was no evidence from randomised controlled trials (RCTs) of the impact of AF screening on stroke and mortality.[33] Two trials of different approaches to AF screening published in 2021 showed promising, but inconclusive results.[46–49] Both recruited much smaller numbers than is planned for Screening for Atrial Fibrillation with ECG to Reduce stroke (SAFER) (approximately 28 000 for STROKESTOP and 6000 for LOOP).[46–49] Neither showed a reduction in ischaemic stroke associated with screening although STROKESTOP reported a reduction in a composite endpoint (ischaemic or haemorrhagic stroke, systemic embolism, death, and hospitalisation for bleeding). As a result in early 2022 the US Preventive Services Task Force did not change its previous recommendation that there was insufficient evidence to determine whether there was greater benefit than harm for ECG screening for AF. Thus, evidence is required from a much larger randomised trial to inform guidelines and national screening body recommendations, a gap that SAFER is intended to fill.

The SAFER trial is a large, pragmatic, open-label, primary care-based RCT which will recruit around 100 000 participants and assess whether screening for AF is effective and cost-effective at reducing stroke and other outcomes.[50] It will randomise participants after consent and will investigate ways to improve implementation of screening. It will use intermittent monitoring via hand-held ECGs which will detect higher-burden AF associated with higher clinical risk than continuous monitoring.[48] It will examine harms as well as benefits of an AF screening programme.[45]

The internal pilot study detailed in this protocol is a cluster RCT recruiting participants who will be followed up during the main trial. The objectives of the internal pilot study are to assess intermediate outcomes such as AF detection rate and anticoagulation rate, reduce uncertainty concerning key parameters for the design, conduct and sample size calculations for the main trial, examine the psychological impact of screening, and investigate ways to optimise the delivery of the AF screening intervention.

## METHODS AND ANALYSIS
### Aim
To inform a decision to proceed to the main trial taking account of key intermediate outcomes (AF detection rate; anticoagulation uptake in screen detected AF), an economic analysis and a revised sample size calculation. Also to assess any psychological impact of screening, and draw lessons for how best to implement screening in the main trial.

### Design
A pragmatic, primary care-based, multicentre, two-parallel arm, open-label, practice-level cluster RCT which aims to recruit 12 600 participants from 36 practices in a 2:1 ratio of usual care (control) to screening (intervention). Participants will be followed up for 12 months for pilot study outcomes, and also for an average of 5 years for main trial outcomes. There will be an embedded process evaluation and qualitative studies, and an economic evaluation. The first practice was randomised on 16 April 2021. Follow-up (for the internal pilot) is scheduled to finish on 30 May 2023.

### Participants and setting
Participating practices will be drawn from a range of UK urban and rural settings, serving patients with a variety of different health and social needs. The vast majority of the UK population is registered with a practice that provides most AF care with referral to secondary care only for more complex cases.[19]

### Eligibility
#### Participants
Broad eligibility criteria have been employed to maximise eternal validity (table 1).

According to guidelines, the vast majority of people aged 70 years or older with AF should be offered anticoagulation.[19] Participants with an existing diagnosis of AF on the practice electronic AF register (which includes both paroxysmal and persistent AF) but who are not being prescribed anticoagulation are included because

**Table 1** Eligibility criteria for participants in the safer pilot and main trial

| Inclusion criteria | Exclusion criteria (as coded on the primary care health record) |
| --- | --- |
| Participant has given valid informed consent | On long-term anticoagulation therapy |
| Aged 70 years or older | On the practice palliative care register |
| | Resident in a nursing or care or residential home |
| | Consented to another trial that will affect participation in SAFER |
| | Non-UK resident |

SAFER, Screening for Atrial Fibrillation with ECG to Reduce stroke.

screening these participants for AF may encourage anti-coagulation use.[42 46 51]

Patients coded as resident in a nursing/care/residential home in the electronic search of patient records will be excluded due to practical difficulties.

Patients taking part in another trial will be excluded if participation in both trials could compromise either trial or affect patients' safety.

### Recruitment

#### Practices

Practice recruitment will be managed by the National Institute for Health and Care Research (NIHR) Clinical Research Network (CRN)—a national network that coordinates and supports research delivery. The CRN will approach practices with information about the trial. Practices will express interest via an online form.

#### Participants

The practice will send approximately 1200 randomly selected eligible patients an invitation pack consisting of a participant information sheet, consent form and Freepost envelope (see online supplemental appendix A to C). In initial practices a negative reply slip will be included in the pack so that reasons for non-participation can be analysed. The exact number invited will vary between practices to achieve recruitment targets based on their characteristics and any associations with recruitment (eg, more invitations to people in more deprived areas).

To facilitate convenience, participants will have the option to return the consent form in a Freepost envelope, or to provide consent online. Reminder invitations, emails, short message service and/or invitation of additional eligible patients may be utilised if response rates are poor.

### Randomisation and allocation

On the day after the recruited and consented participant target number is reached for a practice (350 participants), we will close recruitment and the practice will be randomised, stratified by practice location deprivation

score[52] and prevalence of AF reported in the Quality and Outcomes Framework. No recruitment will take place in a practice once randomisation has occurred.

Randomisation will be implemented using a secure online randomisation system (Sortition[53]) hosted by the University of Oxford Clinical Trials Unit. Practices will be randomised using random permuted blocks within nine strata corresponding to three groups (tertiles) of practice location deprivation score and three groups (tertiles) of practice-level prevalence of AF. The block sizes will be known only to the trial statistician and the randomisation system programmer. All activity on the programme will have an audit trail.

Blinding of allocation to the trial team and to the practices will not be possible.

### Intervention development

The screening intervention was developed with a range of stakeholders that included patient associations, patients, screening policy-makers, general practitioners (GPs) and researchers. The intervention was tested in a feasibility study in 10 practices, which demonstrated that the intervention was feasible and acceptable to participants and practice staff.[54] In this feasibility study, practice staff conducted screening consultations in which participants were instructed how to use the ECG device. In response to the COVID-19 pandemic, a second feasibility study was undertaken in three practices, which showed that a 'remote model' of delivery of the intervention was feasible: participants could be instructed on how to use the ECG device through written instructions and video, and with optional telephone support from the study administrative team. This model ensured a low risk of COVID-19 transmission and reduced workload for primary care. Training of practices was also successfully delivered remotely. This included training on how to manage and discuss results with participants and online anticoagulation training to manage participants in line with current guidelines.[19 55]

The final intervention model is summarised in the logic model in figure 1.

### Screening intervention

Participants in intervention practices will receive an invitation to screening. Those who accept this will receive a call from the study team to arrange home delivery of the single-lead ECG device and written/video instructions, and to offer a subsequent screening consultation if required to provide support. In this consultation, the participants will be guided on use of the device and with the help of test ECG traces, how to produce a trace of acceptable quality.

Participants will undertake 3 weeks of intermittent screening (four 30 s traces each day) as well as when experiencing symptoms (eg, palpitation, dizziness) using the portable Zenicor device (www.zenicor.com). They will transmit their ECG recordings via mobile network to a remote database by pressing a button on the device. If no traces have been received within 10 days, or if more than

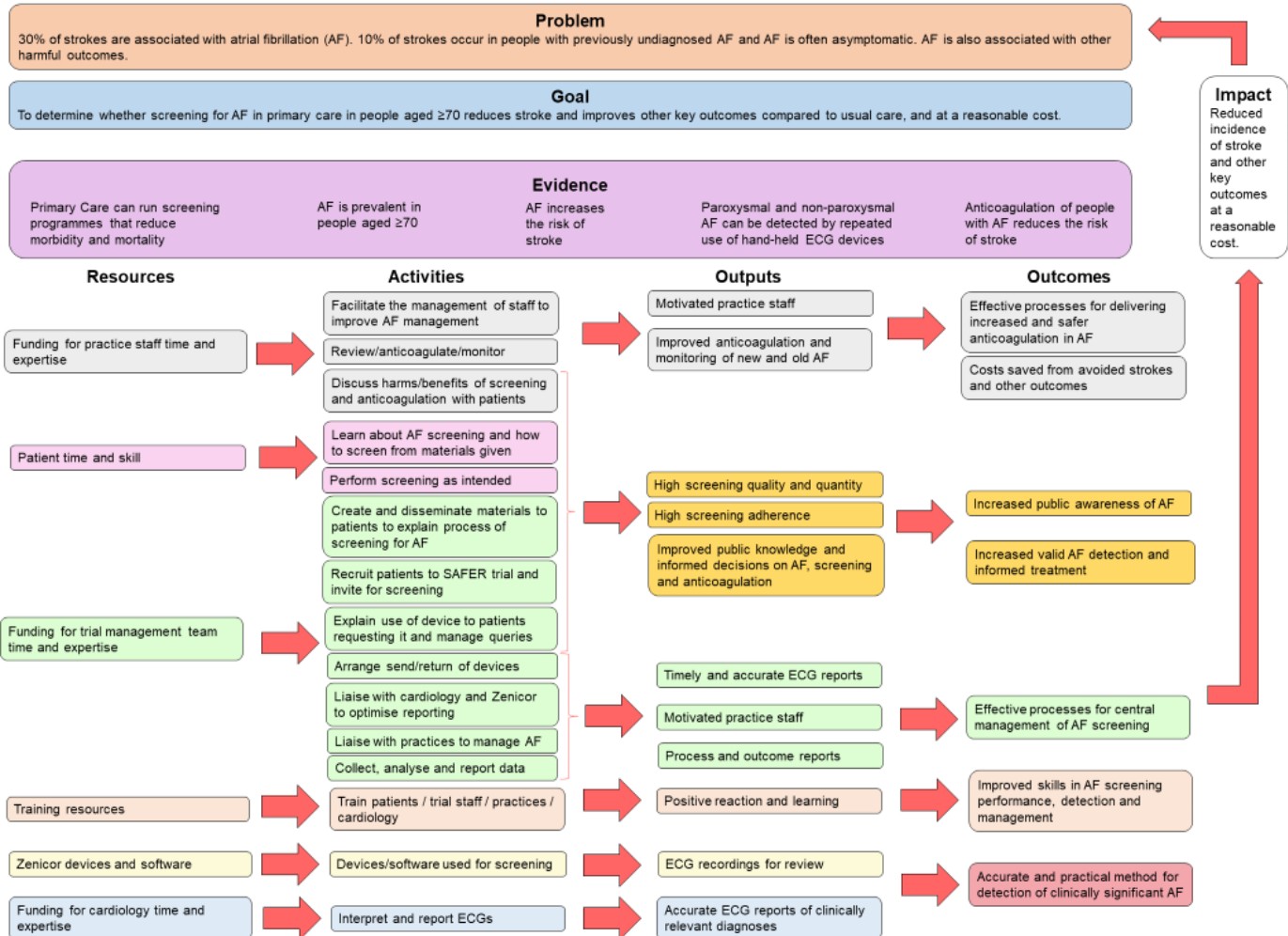

**Figure 1** Logic model of the intervention in the SAFER trial. SAFER, Screening for Atrial Fibrillation with ECG to Reduce stroke.

25% of traces recorded on days 4—10 are tagged by the algorithm as low quality, the trial team will contact the participant to offer further support. We stress to participants both in information sheets and verbally (during the device delivery call) that the ECG device provided should not be used by anyone else.

Participants will be provided with a freepost envelope and asked to return the Zenicor device to the trial team at the end of the screening period.

Practices in the intervention arm are given on-line training on theNational Institute for Health and Care Excellence (NICE) AF guidelines.[19]

### Zenicor device
The screening device being used is the Zenicor hand-held single-lead ECG device. This device is usable in any location, allows repeated ECGs, and can store and transmit multiple ECG traces to a central system for analysis.[21 24] Photoplethysmography[56] and blood pressure machines[57] have not proved accurate enough, and stakeholder discussion deemed patches less practical. The diagnostic model of the Zenicor device, its associated diagnostic algorithms and subsequent cardiologist review have been used

successfully at scale in the STROKESTOP AF screening trial in over 7000 participants.[42 58] The algorithm for detecting AF showed a sensitivity of 98% and specificity of 88%.[59] A photograph of the Zenicor device is shown in figure 2.

### Screening results
A proprietary algorithm will analyse the ECG traces and place a digital flag on ECGs that might show AF. These will be reviewed by a cardiologist or cardiac technician who will determine whether AF or any other important rhythm disturbance is present. If there is uncertainty, the trace will be reviewed by another cardiologist. A confirmatory 12-lead ECG is not required.[60] The cardiologists will create a report with recommendations for the GP. Possible results are shown in table 2.

The trial team will send the screening results to the practice, including copies of relevant ECG traces for positive (AF or other) diagnoses. The GP can access ECG traces and reports for all of their patients freely via the Zenicor web-based system. Practice staff will notify participants of their screening result.

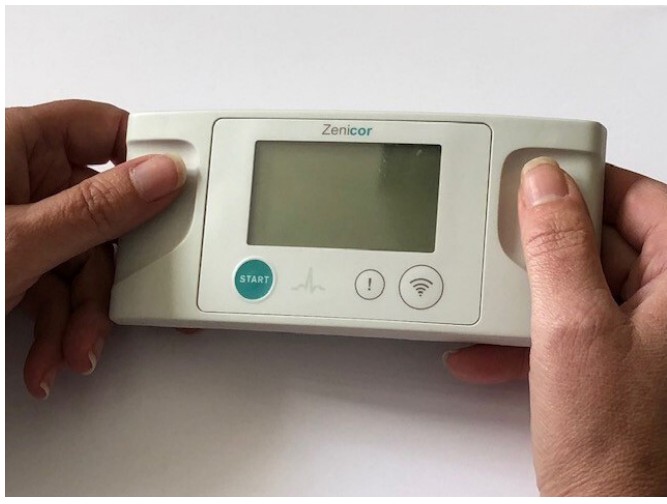

**Figure 2** Zenicor hand-held ECG device used to screen for AF in the SAFER trial. AF, atrial fibrillation; SAFER, Screening for Atrial Fibrillation with ECG to Reduce stroke.

For results 1–3 (table 2), the practices will offer participants a consultation to discuss the result and its appropriate management. GPs are not provided with data on burden of AF, so this will not be considered. See figure 3 for a trial schematic. Practices are monitored to ensure that all patients who are found to have AF are reviewed by their GP.

It is not possible to report results in 'real time'. If participants experience any symptoms, they are advised to seek medical help in the way they usually would, and not wait for the results of the screening (see online supplemental appendix D, screening information leaflet).

| Table 2 | Categories of screening results as reported to general practices in the safer trial |
| --- | --- |
| **Diagnosis** | **Definition** |
| 1. AF ≥30 s | AF is observed for a continuous period of 30 s. Sufficient readable beats (ie, disregarding poor quality sections of an ECG) show AF |
| 2. Cannot exclude AF ≥30 s | Indeterminate result—usually due to poor quality ECG traces |
| 3. Other significant arrhythmia | This may include, but is not limited to:<br>► Second/third degree heart block<br>► Ventricular tachycardia<br>► Supraventricular tachycardia<br>► Any other significant arrhythmia |
| 4. No AF ≥30 s detected | This will include, but is not limited to:<br>► Sinus rhythm<br>► AF <30 s<br>► Bradycardia<br>► Ectopic beats |
| 5. Screening failure | Unable to record any ECGs of sufficient quality for review |
| AF, atrial fibrillation. | |

### Control practices
These will provide usual care, which might involve opportunistic screening.

### Outcomes
Primary and secondary outcomes are shown in box 1. The internal pilot will specifically report on outcomes that are relevant for consideration of continuation of the trial. Participants in the internal pilot study will also be followed up for an average of 5 years for main trial outcomes. The process evaluation during the pilot (protocol to be published separately) will report outcomes to guide the successful delivery of the SAFER main trial and a national-scale AF screening programme.

A random sample of participants stratified by age and sex in both intervention and control arms will be sent questionnaires to assess possible psychological effects of screening. Qualitative work will also contribute to understanding the benefits and harms of screening, and participant experience.

Our definition of newly detected AF is a first AF code recorded within twelve months of randomisation and no AF code in the GP records prior to the date the practice was randomised.

### Sample size
Sample size calculations are based on 350 consented participants from each of the 12 intervention and 24 control practices, and the assumption that 85% of participants per intervention practice will be screened. This will provide a 90% power at 5% significance level to detect a 1.1% absolute difference in the frequency of diagnosis of new AF between intervention and control practices, assuming 3% newly diagnosed AF is detected in screened patients[42] and an intraclass correlation coefficient of 0.001.

Sending the heath questionnaire to 1800 participants will give us 90% power to detect a 4 point difference in the Spielberger questionnaire, assuming 60% respond—the rate achieved in the SAFE trial.[60]

### Data collection
#### Baseline data collection
Baseline data detailed in table 3 will be collected from the GP electronic medical records for all individuals who have consented to participate in the trial.

#### Follow-up data
This section excludes outcomes for the main trial, which will be detailed in the main trial protocol.

*Atrial fibrillation*
1. New diagnoses of AF picked up in both intervention and control practices since screening initiation using GP electronic data.

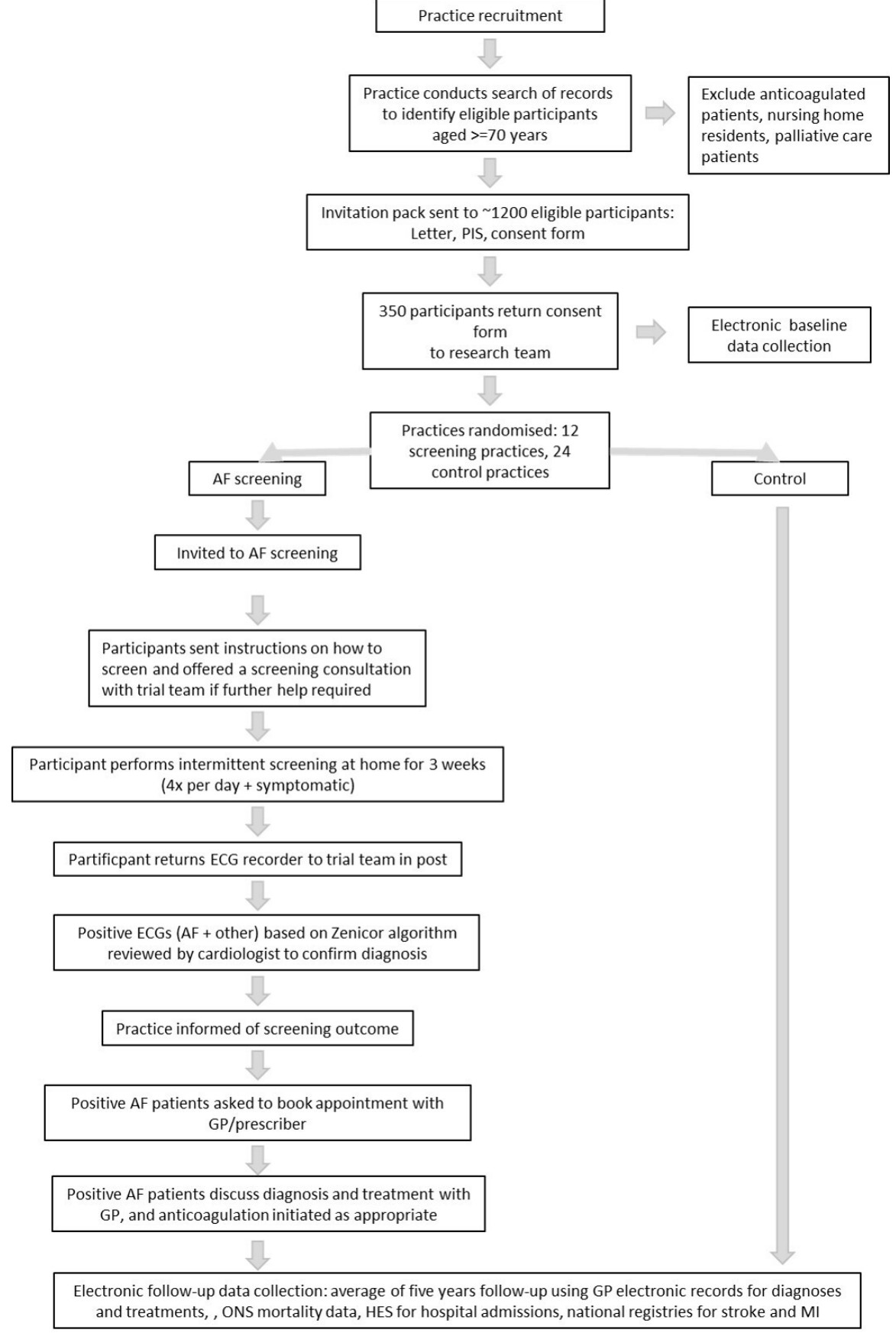

**Figure 3** SAFER trial schematic. AF, atrial fibrillation; GP, general practitioner; PIS, participant information sheet; SAFER, Screening for Atrial Fibrillation with ECG to Reduce stroke; ONS, UK Office for National Statistics; HES, Hospital Episode Statistics; MI, myocardial infarction

**Box 1  Primary and secondary outcomes assessed in the SAFER internal pilot study**

**Primary outcome:**
⇒ Atrial fibrillation
  ⇒ In intervention practices: the number of participants that had AF detected through screening.
  ⇒ In intervention and control practices: the number of newly detected AF patients in intervention practices compared with control practices.

**Secondary outcomes:**
⇒ Uptake of anticoagulation
  ⇒ Proportion of participants with AF detected through screening in intervention practices who were started on anticoagulation.
  ⇒ Number of participants with newly detected AF that were started on anticoagulation in intervention and control practices.
⇒ Parameters to refine the sample size calculation for the main trial (current assumptions in parentheses).
  ⇒ Proportion of consented participants in intervention practices who are screened over the screening period (85%).
  ⇒ Proportion of screened patients in whom newly diagnosed AF is detected (3%).
  ⇒ Proportion of participants with newly diagnosed AF from screening who commence anticoagulation (80%).
  ⇒ Proportion of participants with a known diagnosis of AF that is detected by screening who newly commence anticoagulation, despite previously not being prescribed anticoagulation (55%).

2. Positive diagnoses of AF identified by screening (intervention practices only).

*Uptake of anticoagulation*
1. For patients diagnosed with AF, whether or not they are prescribed anticoagulation (intervention and control practices) using GP electronic data.
2. Initiation of anticoagulation in AF detected through screening programme (intervention practices only).

*Process variables*
1. Whether patients agree to screening (intervention practices only).
2. Whether patients are screened (intervention practices only).

*Psychological outcomes*
The psychological effects of screening and impact on functional status will be assessed by comparing responses to the Spielberger State-Trait Anxiety Inventory (STAI) short form,[61] EQ-5D-5L (EuroQol - 5 Dimensions - 5 Levels)[62] and SF-8 (Short Form-8).[63] Changes in responses over time in both groups will be quantified as well as comparisons of responses according to uptake and outcome of screening. These generic measures may be relatively insensitive to some potential specific impacts of screening for AF, but as they do not include reference to the screening programme they enable comparison between screening and control groups. Furthermore, they facilitate comparisons with screening programmes for other conditions, and with other unrelated health

**Table 3**  Baseline participant data to be collected for the safer pilot study

| Category | Variable collected |
|---|---|
| Demographics | Age |
| | Sex |
| | Ethnicity |
| | Index of Multiple Deprivation based on participant postcode |
| Comorbidities | Atrial fibrillation |
| | Stroke or transient ischaemic attack |
| | Coronary heart disease |
| | Peripheral arterial disease |
| | Heart failure |
| | Hypertension |
| | Diabetes mellitus |
| | Stroke or transient ischaemic attack |
| | Dementia |
| | Depression |
| Clinical scores and indices | $CHA_2DS_2$-VASc score |
| | HAS-BLED score |
| | ORBIT score |
| | Frailty index |
| Other variables | Height |
| | Weight |
| | Alcohol intake |
| | Smoking status |
| | SARS-CoV-2 PCR result |

CHA2DS2-VASc, scoring system to assess the risk of stroke in those with AF; HAS-BLED, scoring system to assess the risk of bleeding in those who take anticoagulation for AF; ORBIT, scoring system to assess the risk of bleeding in those who take anticoagulation for AF

service interventions. The questionnaires will be posted to a random (MS Access RND function) sample of participants (126 per intervention practice and 36 per control practice, matched for age (70–73; 74–77; 78 years and over) and sex in six groups. The target numbers in the intervention arm are raised from our sample size calculation to increase the data available from participants who screen positive. Questionnaires will be posted to the screening group alone at baseline (pre-invitation to screening). Both groups will receive questionnaires after 8 weeks and 6 months.

**Data management**
Data sent from practices to the trial team will be labelled with participant ID number (link-anonymisation), initials and partial date of birth. The local investigator at each site is responsible for case report form integrity. We will offer secure online data capture (including e-consent),

using an established secure system that complies with sponsor security policies (Qualtrics.[64]

ECG traces on the Zenicor system will be labelled with participant ID number, initials and partial date of birth.

Participant questionnaires will be link-anonymised and returned to the trial team by post or online prior to checking and entering.

Participant identifiable data will be stored, handled and processed securely and confidentially, in accordance with sponsor data security policies, UK data laws and ethical guidelines. Access will be restricted to specific members of the trial team. Further information is accessible on the trial website (https://www.safer.phpc.cam.ac.uk/).

## Statistical analysis

Data will be analysed according to Consolidated Standards of Reporting Trials principles and its extension for cluster trials.[65] Outcomes will be analysed using an intention-to-treat principle for primary analysis. However, as both external and internal validity are important in the pilot study, secondary analysis will be conducted according to the per-protocol principle, when necessary and justified.

The proportion of those consented who took up screening, the proportion found to have AF (both new and previously known) and proportion who were antico-agulated will be calculated. The proportion of diagnoses of new AF participants in intervention and control practices and rate of anticoagulation will be compared. Clustering by practices will be accounted for with an adjusted $\chi^2$ test for simple comparisons and mixed effects regression models for covariates.

## Process evaluation and qualitative work

A mixed-methods process evaluation will be conducted to explore how AF screening is delivered and perceived at practice and patient levels. Qualitative work will seek to understand participant experiences of being invited to, and taking part in, the study.

These will contribute to refining the theory of the intervention, which will help provide recommendations for an acceptable and sustainable screening programme at scale.

## Economic analysis

The pilot data will be used to update a published model, composed of a decision tree followed by a Markov model.[23] The purpose of this model is to confirm that it is plausible that screening might be cost-effective using the parameters obtained in this pilot trial. All patients entering the decision tree will incur an invitation cost and the test cost will be applied to those patients who accept screening. Screen-negative patients will not accrue any additional costs and quality-adjusted life-years (QALYs). The remaining patients are true positive and, thus, will enter the Markov model. This model will simulate their survival trajectories accounting for their condition and, their lifetime costs and QALYs, which will be discounted at a 3.5% annual rate and half-cycle corrected.

The costs needed to implement the screening programme will be calculated using a microcosting approach to include all the relevant costs, such as the invitation cost and cost due to the device use (eg, shipment of the device and the training to use the device).[66] Where needed, the costs incurred by the National Health Service (NHS) will be updated using the most recent available data, such as the British National Formulary for the cost of anticoagulant therapies.[67]

The model will be employed to perform a probabilistic analysis and compute the total costs and QALYs. The differences in costs and QALYs between the SAFER intervention and usual care will be calculated and combined to obtain the incremental cost-effectiveness ratio. Likewise, the expected value of perfect information (EVPI) will be calculated by assuming that the value of one QALY is equal to £20 000, which reflects the cost-effectiveness threshold used by the NICE. Then, the EVPI will be projected to the national level considering the eligible population for the screening and assuming that the screening programme will be provided for the next 10 years.[68]

## Management and oversight

The University of Cambridge and NHS Cambridgeshire & Peterborough Clinical Commissioning Group are cosponsors. The trial management group (consisting of the chief investigator and researchers from each group) and the programme steering committee (PSC), which has an independent chair and four independent members, will appraise data and decide on continuation and course of the study in consultation with the NIHR. An active risk register has been compiled in consultation with the funder and sponsors, and will be monitored and updated throughout.

## Patient and public involvement

The SAFER programme has been guided since inception by patient and public representatives who participate in all-investigator meetings. Trudie Lobban, chief executive and founder of the Atrial Fibrillation Association (AFA), has been involved in the development of the research from the outset as a patient and public involvement (PPI) member. The AFA represents over 64 000 people with AF.

Additional PPI members have been recruited independently of the AFA. Many are in the age range for AF screening; some of them either have AF or have a partner with AF. The PSC has an independent lay member who is a stroke survivor.

The PPI members are consulted throughout the trial on all aspects of the research, including: possible psychological harms of screening; participant-facing documents; how to approach participants; instructing participants on trials and screening procedures; web-based materials and qualitative data-collection material. The AFA will help with dissemination of the findings through its website and members.

## ETHICS AND DISSEMINATION

### Ethics

#### Ethical approval

The SAFER pilot trial has received a favourable ethical opinion from the London—Central NHS Research Ethics Committee (19/LO/1597) and the Confidentiality Advisory Group (19/CAG/0226). Modifications of the full protocol are detailed in amendments. Important modifications will be communicated to the sponsors, funder, collaborators, practices, participants, trial registries and disseminators as relevant.

### Consent

Participants will be required to provide valid written informed consent, either via post or online. Consented participants from screening practices will be approached with an offer of AF screening.

### Dissemination

The study will generate peer-reviewed publications to disseminate to academics, health professionals, policy-makers, patient organisations and the print and electronic media. After publication, data may be available to others according to data sharing agreements in compliance with the funder and sponsor policies. Summary documents will be made available to participants at the end of the study. PPI groups and media engagement will help disseminate findings. Accessible reports will be generated for national screening committees, commissioners and other decision-makers. Funders' reports will be submitted in accordance with their policies.

**Author affiliations**
[1]Strangeways Research Laboratory, Primary Care Unit, Department of Public Health and Primary Care, University of Cambridge, Cambridge, UK
[2]THIS (The Healthcare Improvement Studies) Institute, University of Cambridge, Cambridge, UK
[3]Bristol Medical School, University of Bristol, Bristol, UK
[4]Heart Arrhythmia Alliance, Chipping Norton, UK
[5]Primary Care Population Sciences and Medical Education, University of Southampton School, Southampton, UK
[6]Department of Health Sciences, George Davies Centre, University of Leicester, Leicester, UK
[7]Department of Public Health and Primary Care, University of Cambridge, Cambridge, UK
[8]Liverpool Centre for Cardiovascular Science, University of Liverpool, Liverpool John Moores University and Liverpool Heart & Chest Hospital, Liverpool, UK
[9]Department of Clinical Medicine, Aalborg University, Aalborg, Denmark
[10]Guy's & St Thomas' NHS Foundation Trust, Royal Brompton Hospital, London, UK
[11]Faculty of Life Sciences and Medicine, Kings College London, London, UK
[12]Faculty of Medicine, National Heart and Lung Institute, Imperial College London, London, UK
[13]Health Sciences, Warwick Medical School, University of Warwick, Coventry, UK
[14]Heart research Institute, Charles Perkins Centre, The University of Sydney, Sydney, New South Wales, Australia
[15]MRC Epidemiology Unit, School of Clinical Medicine, University of Cambridge, Cambridge, UK
[16]Nuffield Department of Primary Care Health Sciences, University of Oxford, Oxford, UK

**Contributors** JM is the guarantor. RNM drafted the manuscript. KW and AD are coordinating, planning gaining ethical approval, conduct and helping design the study. JM, JB, NA, DE, TL, ML, MJS, GL, MRC, DAF, BF, SJG, SS, FDRH and RJM undertook design, planning and oversaw conduct of the study. RNM, SH, AP, JB, RJ and NA designed the process evaluation and qualitative studies. JL designed the collection and analysis of some of the pilot outcome data collection. TL is a PPI representative that has informed design, outcomes and dissemination plan. SM, FF and HT designed the economic evaluation and will oversee its conduct. MJS and SK designed the statistical analysis and will oversee its conduct. The SAFER author group contributed to planning and design of study, applying for funding, oversaw conduct, and writing of the protocol for the ethical approval. All authors reviewed and had the option to edit the final manuscript.

**Funding** The SAFER pilot study and main trial are funded by the National Institute for Health and Care Research (NIHR) (Programme Grants for Applied Research Programme (Reference Number RP-PG0217-20007)). The SAFER feasibility study is funded by the NIHR (School for Primary Care Research (SPCR-2014-10043, project 410)). SAFER is a contributor to/partner of AFFECT-EU receiving funding from the European Union's Horizon 2020 research and innovation Programme under grant agreement NO. 847770. RM and JL are supported by the Wellcome Trust as part of the Wellcome Trust PhD Programme for Primary Care Clinicians (grant number 203921/Z/16/Z). JB, SH and AP are based in The Healthcare Improvement Studies Institute (THIS Institute), University of Cambridge. THIS Institute is supported by the Health Foundation, an independent charity committed to bringing about better health and healthcare for people in the UK. JB is supported by the Health Foundation's grant to the University of Cambridge for The Healthcare Improvement Studies (THIS) Institute (RG88620). JM and FDRH are NIHR Senior Investigators. FDRH acknowledges support from NIHR ARC OTV and Oxford BRC (OUT). RJM is an NIHR Senior Investigator and acknowledges support from NIHR ARC OTV. NA is supported by a Health Foundation Improvement Science Fellowship and also by the NIHR Applied Research Collaboration East Midlands (ARC EM). RJ is an NIHR-funded Academic Clinical Lecturer. The University of Cambridge has received salary support in respect of SJG from the NHS in the East of England through the Clinical Academic Reserve. BF received funding from the Medical Research Future Fund International Clinical Trial Collaboration Grant to perform SAFER-AUS as part of SAFER, and a NSW Health Senior Researcher Cardiovascular Grant for work in AF.

**Disclaimer** All the funders and sponsors had no involvement in the development of this protocol and will have no involvement in any aspect of the study itself. The views expressed are those of the author(s) and not necessarily those of the NHS, the Wellcome Trust, the NIHR or the UK Department of Health and Social Care

**Competing interests** JM has performed consultancy work for BMS/Pfizer and Omron. FDRH reports occasional consultancy for BMS/Pfizer, Bayer and BI over the past 5 years. NA is a member of the UK National Screening Committee's Adult Reference Group. MS is a full-time employee of AstraZeneca. MRC reports consultancy for AstraZeneca, Abbott, Medtronic, Bayer, Novartis, Boehringer-Ingelheim-Lilly Alliance, Servier & Pfizer over the past 5 years. RMc's employer the University of Oxford receives consultancy and licencing payments from Omron and Sensyne for BP telemonitoring interventions. GYHL: Consultant and speaker for BMS/Pfizer, Boehringer Ingelheim and Daiichi-Sankyo. No fees are received personally. SJG has received honoraria from Astra Zeneca for lectures at postgraduate educational meetings for primary care teams about type 2 diabetes. BF has received speaker fees, honoraria, and non-financial support from the BMS and Pfizer Alliance; grants to the Institution for investigator-initiated studies from the BMS and Pfizer Alliance; and loan devices for investigatorinitiated studies from Alivecor: all were unrelated to the present study but related to screening for AF.

**Patient and public involvement** Patients and/or the public were involved in the design, or conduct, or reporting, or dissemination plans of this research. Refer to the Methods section for further details.

**Patient consent for publication** Consent obtained directly from patient(s)

**Provenance and peer review** Not commissioned; externally peer reviewed.

**ORCID iDs**
Rakesh Narendra Modi http://orcid.org/0000-0001-9651-6690
Sarah Hoare http://orcid.org/0000-0002-8933-217X
Mark Lown http://orcid.org/0000-0001-8309-568X
Stephen Morris http://orcid.org/0000-0002-5828-3563
Gregory Lip http://orcid.org/0000-0002-7566-1626
Natalie Armstrong http://orcid.org/0000-0003-4046-0119
Martin R Cowie http://orcid.org/0000-0001-7457-2552
Ben Freedman http://orcid.org/0000-0002-3809-2911
Richard J McManus http://orcid.org/0000-0003-3638-028X

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
