## [Reviewer comments · BMJ Open]

ARTICLE DETAILS

TITLE (PROVISIONAL)	A cluster randomised controlled trial of screening for atrial fibrillation in people aged 70 years and over to reduce stroke: protocol for the pilot study for the SAFER trial
AUTHORS	Williams, Kate; Modi, Rakesh; Dymond, Andrew; Hoare, Sarah; Powell, Alison; Burt, Jenni; Edwards, Duncan; Lund, Jenny; Johnson, Rachel; Lobban, Trudie; Lown, Mark; Sweeting, Michael J.; Thom, H; Kaptoge, Stephen; Fusco, Francesco; Morris, Stephen; Lip, Gregory; Armstrong, Natalie; Cowie, Martin; Fitzmaurice, David; Freedman, Ben; Griffin, Simon; Sutton, Stephen; Hobbs, Richard; McManus, Richard; Mant, Jonathan; SAFER Authorship Group, The

VERSION 1 – REVIEW

REVIEWER	Wong, Kam Cheong The University of Sydney Faculty of Medicine and Health, Westmead Applied Research Centre
REVIEW RETURNED	12-Jun-2022

GENERAL COMMENTS	 1. “Participants with an existing diagnosis of AF but who are not being prescribed anticoagulation are included because screening these participants for AF may encourage anticoagulation use”: Are these patients with paroxysmal AF or persistent AF? Did they have AF at enrollment? Is this a subgroup analysis examining a different outcome, i.e. ‘anticoagulation use’ rather than ‘AF detection rate’? This should be delineated to avoid contamination of the overall AF detection rate. 2. “Recruitment demographics will be monitored. If certain populations, e.g. the very elderly, are underrepresented, they will be over-sampled”: What are the definitions of “very elderly”, “underrepresented”, and “over-sampled”? 3. Participants are selected by searching the GP’s electronic medical records. Is the search limited to patients who have visited the GP practices within the last X years? What is ‘X’? 4. Participants are asked to record four ECG traces daily for three weeks using a single-lead handheld ECG device (Zenicor). With reference to Figure 3, the ECG traces interpretation seems to occur after participants return the device to the trial team. However, on page ‘10 of 51’, the authors report that the “Participants will be followed up immediately”. What does “immediately” mean? It is unclear will the participants receive real-time notification (e.g. on the same day) when atrial fibrillation or other significant ECG abnormalities are detected? 5. “The cardiologists will create a report with recommendations for the GP. Possible results are shown in Table 2.” Third-degree heart block and ventricular tachycardia are included in Table 2. These are critical abnormalities that require urgent medical attention. How soon will the GP be notified of these critical abnormalities?
--

	6. Will the 'level of agreement between the device's automatic interpretation and clinicians' interpretation' be assessed and reported? 7. "If no traces have been received, or if more than 25% of traces are tagged by the algorithm as low quality, the trial team will contact the participant": No traces have been received for how many days? Please elaborate on how '25% low-quality trace' is computed? 8. "An active risk register has been compiled in consultation with the funder and sponsors and will be monitored and updated throughout": What is the risk mitigation plan for this risk register? Can it be included as an appendix? Thank you.
--	--

REVIEWER	Gruwez, Henri Ziekenhuis Oost-Limburg, Cardiology
REVIEW RETURNED	20-Jun-2022

GENERAL COMMENTS	They authors describe a study protocol to asses feasibility and refine the implementation of screening for atrial fibrillation (AF) for a consecutive larger trial that aims to determine whether screening for AF is effective at reducing risk of stroke. I would like to congratulate the authors on their manuscript. I really enjoyed reading it. The question whether screening for AF is affective at reducing stroke is relevant. The study protocol as described will provide results that may help to answer this question. These are my comments:  - They authors estimate the proportion of participants with newly diagnosed AF from screening who commence anticoagulation will be 80%. This is a potential weakness of the study. What is the study protocol for the general practitioner (GP) following a positive screening? Without a strict protocol the study is prone to variations between GP's and GP practices. Why not protocolize it and strive for 100%? - It remains unclear how patients in the control group will be screened. Will they receive opportunistic screening for AF as guideline recommended? How will this be performed? - 'The trial team will send the screening results to the practice, including copies of relevant ECG traces for positive (AF or other) diagnoses.' Some measurements may be performed by someone other than the patient with the device (eg spouse). Will any additional information be provided to the GP practices to cope with such issue? Also, again, without directive to the GP practice some may handle one positive measurement as a threshold to start treatment, others may require two, etc. In my opinion this can be avoided by adding a protocol on screen positive measurements with the screening results sent to the GP practices. - The Zenicor device will be used with cardiologist review. It remains unclear whether all measurements be reviewed or only AF positive measurements will be reviewed?
---

VERSION 1 – AUTHOR RESPONSE

Reviewer 1	Response
1. "Participants with an existing diagnosis of AF but who are not being prescribed	Existing diagnosis includes both paroxysmal and persistent AF. Such patients will have a diagnosis

anticoagulation are included because screening these participants for AF may encourage anticoagulation use”: Are these patients with paroxysmal AF or persistent AF? Did they have AF at enrollment? Is this a subgroup analysis examining a different outcome, i.e. ‘anticoagulation use’ rather than ‘AF detection rate’? This should be delineated to avoid contamination of the overall AF detection rate.	of AF on their GP electronic records at enrolment. In box 1 we list anticoagulation as a secondary outcome. To avoid contamination of the overall AF detection rate, our primary outcome (as shown in box 1) includes comparison of newly detected AF patients in intervention practices as compared with control practices. To clarify this in the manuscript, we have made the following changes: -Participants with an existing diagnosis of AF on the practice electronic AF register (which includes both paroxysmal and persistent AF) but who are not being prescribed... -In the section headed ‘Outcomes’ we have inserted: Our definition of newly detected AF is a first AF code recorded within twelve months of randomisation and no AF code in the GP records prior to the date the practice was randomised.
2. “Recruitment demographics will be monitored. If certain populations, e.g. the very elderly, are underrepresented, they will be over-sampled”: What are the definitions of “very elderly”, “underrepresented”, and “over-sampled”?	This sentence was in our ethics application to enable us to change the sampling strategy in the internal pilot if we needed to. We did not define what we meant by ‘very elderly’ or ‘under-represented’. We accept that the way the text is worded raises ambiguities that we cannot in truth address. Given that we have not altered our sampling strategy we have deleted this text: -Recruitment demographics will be monitored. If certain populations, e.g. the very elderly, are underrepresented, they will be over-sampled.
3. Participants are selected by searching the GP’s electronic medical records. Is the search limited to patients who have visited the GP practices within the last X years? What is ‘X’?	In the UK, the vast majority of the UK population is registered with a practice. Registration with a practice does not require the participant to visit the practice. Similarly, visiting the practice was not a requirement for patient selection. We have not altered the current text which states: -The vast majority of the UK population is registered with a practice that provides most AF care...
4. Participants are asked to record four ECG traces daily for three weeks using a single-lead handheld ECG device (Zenicor). With reference to Figure 3, the ECG traces interpretation seems to occur after participants return the device to the trial team. However, on page ‘10 of 51’, the authors report that the “Participants will be followed up immediately”. What does “immediately” mean? It is unclear will the participants receive real-time notification (e.g. on the same day) when atrial fibrillation or other significant ECG abnormalities are detected?	The flow chart is correct. There is no possibility for real time notification in this study (but see response to point 5 below), since the ECGs may not be read for several weeks after they were recorded. The quoted text about immediate follow up is distinguishing the immediate pilot trial outcomes (AF detection) from the delayed main trial outcomes (stroke and other clinical events). On reflection, the use of the term immediate was not helpful. We have amended the text so that it now states: - Participants will be followed up immediately for 12 months for pilot study outcomes, and also for an average of five years for main trial outcomes.
5. “The cardiologists will create a report with recommendations for the GP. Possible results are shown in Table 2.” Third-degree heart block and ventricular tachycardia are included in Table 2. These are critical abnormalities that	As noted above, real time reporting is not possible. The participants are made aware of this when they receive the invitation for screening, and that should they have any symptoms, they should seek medical help in the usual way. There is ‘expedited’ reporting in that if a cardiologist sees a

require urgent medical attention. How soon will the GP be notified of these critical abnormalities?	trace that is life threatening, they will alert the study team immediately, and we will contact the practice. But this is still likely to be a few weeks after the rhythm was recorded. We have added the following text to the section 'screening results' to clarify this: -It is not possible to report results in 'real time'. If participants experience any symptoms, they are advised to seek medical help in the way they usually would, and not wait for the results of the screening. We have also added an additional appendix, our screening information leaflet: -see appendix D, Screening Information Leaflet
6. Will the 'level of agreement between the device's automatic interpretation and clinicians' interpretation' be assessed and reported?	Not as part of the internal pilot trial analysis, as such analyses for this device have already been published in the literature (Svennberg E, Stridh M, Engdahl J, Al-Khalili F, Friberg L, Frykman V, Rosenqvist M. Safe automatic one-lead electrocardiogram analysis in screening for atrial fibrillation. EP Europace. 2016;19(9):1449-53). We have added this reference to the paper where we refer to the device: - The diagnostic model of the Zenicor device, its associated diagnostic algorithms, and subsequent cardiologist review have been used successfully at scale in the STROKESTOP AF screening trial in over 7000 participants.(42,58) The algorithm for detecting AF and showed a sensitivity of 98% and specificity of 9288%.(59)
7. "If no traces have been received, or if more than 25% of traces are tagged by the algorithm as low quality, the trial team will contact the participant": No traces have been received for how many days? Please elaborate on how '25% low-quality trace' is computed?	We have clarified in the text: -If no traces have been received within 10 days, or if more than 25% of traces recorded on days 4 to 10 are tagged by the algorithm as low quality, the trial team will contact the participant...
8. "An active risk register has been compiled in consultation with the funder and sponsors and will be monitored and updated throughout": What is the risk mitigation plan for this risk register? Can it be included as an appendix?	We stated this as part of our summary of management and oversight. The risk register is constantly being updated and changed as we take actions to mitigate some risks, and as new risks occur. We regard this as an internal document for the view of the Trial Management Group and the Independent Trial Steering Committee. It includes potentially sensitive information, so we do not plan to put it in the public domain.
Reviewer 2	
9.. They authors estimate the proportion of participants with newly diagnosed AF from screening who commence anticoagulation will be 80%. This is a potential weakness of the study. What is the study protocol for the general practitioner (GP) following a positive screening? Without a strict protocol the study is prone to variations between GP's and GP practices. Why not protocolize it and strive for 100%?	We do not strive for 100%. This is a pragmatic trial where we want to see whether screening is effective in real world conditions. Furthermore, the NICE guidelines under which the GPs operate advise: 'offer anticoagulation (if) CHA2DS2-VASc score of 2 or above, taking into account the risk of bleeding, and: 'consider' anticoagulation (if) CHA2DS2-VASc score is 1. (This will apply to men aged 70-74 without any risk factors). For some patients, the risks of treatment will be felt to outweigh the benefits. That said, we do strive to have as high uptake of anticoagulation as

	possible. With this in mind, all intervention practices received training on the NICE guidelines, and the study centre ensured that all patients who were found to have AF were reviewed by their GP. We have added the following:  -under the section screening intervention: Practices in the intervention arm are given on-line training on the NICE AF guidelines.(19) -under the section screening results: Practices are monitored to ensure that all patients who are found to have AF are reviewed by their GP.
10. It remains unclear how patients in the control group will be screened. Will they receive opportunistic screening for AF as guideline recommended? How will this be performed?	We have added a new section (immediately after 'screening results') -Control practices These will provide usual care, which might involve opportunistic screening.
11. 'The trial team will send the screening results to the practice, including copies of relevant ECG traces for positive (AF or other) diagnoses.' Some measurements may be performed by someone other than the patient with the device (eg spouse). Will any additional information be provided to the GP practices to cope with such issue? Also, again, without directive to the GP practice some may handle one positive measurement as a threshold to start treatment, others may require two, etc. In my opinion this can be avoided by adding a protocol on screen positive measurements with the screening results sent to the GP practices.	The possibility of a measurement being performed by someone other than the patient: We address this in several ways. Firstly, the written materials that participants receive emphasise that the device must not be used by anyone else. This is further emphasised when participants are contacted by telephone to arrange delivery of the ECG device. Secondly, if there are two participants from the same household, they are each sent a device at separate time points. We have added the following text:  - under 'screening intervention': We stress to participants both in information sheets and verbally (during the device delivery call) that the ECG device provided should not be used by anyone else. Threshold to start treatment: The GPs are not sent all the positive ECGs, just a sample. They are not informed how many ECGs are positive. In their training and in written information sent to GPs they are advised to act on a single ECG showing AF. We have clarified as follows:  - the practices will offer participants a consultation to discuss the result and its appropriate management. GPs are not provided with data on burden of AF, so this will not be considered.
12. The Zenicor device will be used with cardiologist review. It remains unclear whether all measurements be reviewed or only AF positive measurements will be reviewed?	ECGs that are not flagged by the algorithm are not reviewed by the cardiologists. This is implicit in the following:  - A proprietary algorithm will analyse the ECG traces and place a digital flag on ECGs that might show AF. These will be reviewed by a cardiologist or cardiac technician who will determine whether AF or any other important rhythm disturbance is present. If there is uncertainty, the trace will be reviewed by another cardiologist. A confirmatory 12-lead ECG is not required.
Editor Comments	
1. Please ensure that the information provided in your protocol article is	Thank you. We have reviewed the protocol article and the registry for inconsistencies.

consistent with that included in the trial registry. For example, Exclusion criteria and sample size. Please update the manuscript and/or trial registry accordingly.	With regards to exclusion criteria the BMJ Open article is correct and we have updated the registry accordingly. With regards to sample size, the BMJ Open article refers to 36 practices, which was our original intention. It was always our intention to carry on recruiting practices (to the main trial) after the pilot trial had finished, which is why we recruited 39 practices to the cluster randomised element. However, a decision was made to conduct the main trial as an individually randomised study. Therefore the ISCRTN registry refers to 39 practices since we will include all 39 in our write up of the internal pilot. Given that the protocol submitted to the BMJ Open aims to describe what we intended to do, we have left these numbers unchanged.
2. - Please ensure that your protocol reports all outcome measures for your trial and ensure that the primary and secondary outcome measures are consistent between your protocol article and the trial registry	The BMJ Open protocol article reports the outcomes of the internal pilot trial. The registry, while it refers to the internal pilot, only reports the outcomes for the main trial. This explains the discrepancy.
3. Please include the planned start and end dates for the study in the methods section	We have deleted the following from the introduction: ... starting in March 2021 is a cluster RCT in 36 clusters (general practices), recruiting 12,600 participants who will be followed up during the main trial. We have inserted the following under design (in methods): The first practice was randomised on 16th April 2021. Follow up (for the internal pilot) is scheduled to finish on 30th May 2023.

VERSION 2 – REVIEW

REVIEWER	Wong, Kam Cheong The University of Sydney Faculty of Medicine and Health, Westmead Applied Research Centre
REVIEW RETURNED	28-Jul-2022
GENERAL COMMENTS	Thank you for addressing the reviewers' comments.
REVIEWER	Gruwez, Henri Ziekenhuis Oost-Limburg, Cardiology
REVIEW RETURNED	16-Aug-2022
GENERAL COMMENTS	They authors describe a study protocol to asses feasibility and refine the implementation of screening for atrial fibrillation (AF) for a consecutive larger trial that aims to determine whether screening for

	AF is effective at reducing risk of stroke. I would like to congratulate the authors on their manuscript. I really enjoyed reading it. The question whether screening for AF is affective at reducing stroke is relevant. The study protocol as described will provide results that may help to answer this question. My comments were countered with sufficient additional information provided by the authors in the latest manuscript. I look forward to see this published.
--	---